# Cultural and Ethical Barriers to Cancer Treatment in Nursing Homes and Educational Strategies: A Scoping Review

**DOI:** 10.3390/cancers13143514

**Published:** 2021-07-14

**Authors:** Cynthia Filteau, Arnaud Simeone, Christine Ravot, David Dayde, Claire Falandry

**Affiliations:** 1Service de Gériatrie, Centre Hospitalier Lyon Sud, Hospices Civils de Lyon, 69495 Pierre-Bénite, France; christine.ravot@chu-lyon.fr (C.R.); claire.falandry@chu-lyon.fr (C.F.); 2Département de Gériatrie, Hôpital Maisonneuve-Rosemont, Université de Montréal, Montréal, QC H1T 2M4, Canada; 3Université Lumière-Groupe de Recherche en Psychologie Sociale (UR GRePS) Institut de Psychologie, 69676 Bron, France; arnaud.simeone@univ-lyon2.fr; 4Plateforme de Recherche de l’Institut de Cancérologie des Hospices Civils de Lyon, Centre Hospitalier Lyon Sud, Hospices Civils de Lyon, 69495 Pierre-Bénite, France; David.DAYDE@chu-lyon.fr; 5Laboratoire CarMeN, INSERM, INRAE, Université Claude Bernard Lyon-1, 69600 Oullins, France

**Keywords:** cancer, barriers to treatment, nursing homes, education

## Abstract

**Simple Summary:**

The increasing prevalence of cancer in nursing homes may be explained by the increased incidence of cancer associated with older age combined with treatment effectiveness leading to chronicity of cancer and functional decline related or not to cancer itself or its treatment. It represents a major challenge for paramedical staff poorly equipped to manage active cancer treatments and may be influenced by unconscious agism due to misconception of cancer care in the elderly. The ONCOPAD project aims to develop information and training programs for paramedical staff in nursing homes to demystify the management of cancer in the elderly and improve the quality of care outside the hospital and the quality of life of patients at each stage. Considering the need to adapt the program, a systematic scoping review of the current data was carried out to map educational strategies as well as cultural and ethical barriers associated with the treatment of cancer in nursing homes.

**Abstract:**

(1) Background: The aging of the population, the increase in the incidence of cancer with age, and effective chronic oncological treatments all lead to an increased prevalence of cancer in nursing homes. The aim of the present study was to map the cultural and ethical barriers associated with the treatment of cancer and educational strategies in this setting. (2) Methods: A systematic scoping review was conducted until April 2021 in MEDLINE, Embase, and CINAHL. All articles assessing continuum of care, paramedical education, and continuing education in the context of older cancer patients in nursing homes were reviewed. (3) Results: A total of 666 articles were analyzed, of which 65 studies were included. Many factors interfering with the decision to investigate and treat, leading to late- or unstaged disease, palliative-oriented care instead of curative, and a higher risk of unjustified transfers to acute care settings, were identified. The educational strategies explored in this context were generally based on training programs. (4) Conclusions: These results will allow the co-construction of educational tools intended to develop knowledge and skills to improve diagnostic and therapeutic decision-making, the consistency of care, and, ultimately, the quality of life of older cancer patients in nursing homes.

## 1. Introduction

Western societies and health systems must adapt continually to the ethical and organizational challenges associated with an aging population [1]. Due to the exponential increase in both cancer and dependence in the very elderly, management of cancer in nursing homes is becoming a major challenge. At the same time, the incidence of cancer increases with age, particularly at around 80 years, and it is becoming a chronic disease due to therapeutic innovations that increase the effectiveness and tolerability of oncological treatment [1,2,3]. This parallel increase in both dependence and cancer in an older population induces ethical debates on how to balance the benefits of specific cancer treatments—evaluated, for example, through quality of life—and their potential harms—linked, for example, to excessive toxicity despite the recent improvement in the adaptation of treatment to the elderly [4]. It also leads to questions about how to validate treatment decisions in cognitively impaired populations and to collect consent from the patient and his/her family [5,6,7]. In this context, the cancer continuum of care may be influenced by organizational, cultural, and ethical barriers associated with the lack of knowledge of the healthcare team, including nurses of nursing homes.

The ONCOPAD project (Programme d’accompagnement à la gestion des parcours ONCOlogiques chez les Personnes Agées Dépendantes) aims to develop information and training programs for paramedical staff in nursing homes to apprehend different clinical oncological situations and improve the care and quality of life of patients at each stage. Due to their everyday proximity to nursing home residents, paramedical professionals constitute a key member in the continuum of care and the main target of the project.

Considering the need for tailoring such an information and training program dedicated to nursing home paramedical professionals, a systematic analysis of current evidence was performed according to scoping review standards. A systematic scoping review aims to map evidence with a systematic approach on a topic and identify the main concepts that are heterogeneous in methods or discipline, or to identify gaps in the literature [8]. The topic of this scoping review concerns cultural and ethical barriers associated with the treatment of cancer in nursing homes and educational strategies for continuing education aimed at postgraduate oncology nurses.

## 2. Materials and Methods

### 2.1. Study Selection

Considering the need to explore both the continuum of cancer care in nursing homes, from diagnosis to palliative care, and post-graduation educational tools, the selection criteria were broad. To be included in the review, studies needed to focus on the different stages of the oncology continuum of care in nursing homes and educational strategies for continuing education targeting nurses. Inclusion criteria were original, qualitative, quantitative, and mixed-method studies published in 2000 or thereafter, written in English or French language. Exclusion criteria were non-original articles (other reviews of the literature, case studies, abstracts from symposia), publication date before 2000, and language other than English or French, as well as screening or specific cancer as the topic. To identify relevant papers, the electronic databases Embase, CINAHL, and MEDLINE were searched until April 2021. Full details of the search strategy for all databases are given in the Appendix A. A snowballing procedure was used to examine the references cited in the articles retrieved through the systematic search. Grey literature was also screened via Open Gray. The final search results were exported into Zotero (v5.0.94), and duplicates were removed. Two reviewers (Cy.F., Cl.F.) independently searched titles/abstracts for eligibility. Any disagreement between reviewers was resolved through discussion and a final agreement was reached. The reviewers assessed the full text of the articles retained after the title/abstract screening. Considering the exclusion of systematic scoping reviews from PROSPERO registration [9], this systematic analysis was not registered on any platform. The methodology used was based on the Preferred Reporting Items for Systematic Reviews and Meta-Analyses Extension for Scoping Reviews (PRISMA-ScR) checklist (Appendix A) [10].

### 2.2. Data Extraction

The collected variables included study design, population enrolled, setting with types of facilities and country, inclusion criteria, intervention, main outcome measures (when applicable, geriatric and oncology issues, functional status, comorbidities, cognitive status, and mortality), summary of results, and sources of bias.

### 2.3. Definitions

Nursing homes: service providing 24-h care to individuals who have complex care needs that can no longer be met in the community.

Assisted living facilities: similar 24-h care by a registered nurse but in a more home-like environment, including apartments and group homes [11].

Palliative care: comprehensive comfort care for patients with a terminal illness that do not respond to attempts to cure or slow the progression of the disease [12].

## 3. Results

### 3.1. Database Search

A total of 666 records were identified, including five references from other sources. After removal of duplicates, 605 records were eligible for selection, of which 83 corresponded to the inclusion criteria. All records screened (*n* = 605) are listed in the Appendix A. After the full-text review, 18 articles were excluded, and 65 studies were included in the analysis (Figure 1).

Due to the broad content of this scoping review, seven different categories emerged after reading the articles reporting the 65 studies: testing (cancer screening in nursing homes and decision-making process surrounding further investigations for cancer diagnosis), treatment (specificities of older cancer patients and nursing home residents, cancer management in nursing homes, and oncological treatment outcomes in this age group), symptoms (including the impact of cognitive impairment and pain management on these), advanced care planning (factors associated with do-not-resuscitate decisions of nursing home residents, disease trajectories, and implication of nurses in these decisions), end of life (mortality in nursing homes and palliative care), nursing home care (philosophy of care and transitions between this setting and hospitals), and education (strategies and facilitators for continuing education in nursing homes). The characteristics of the studies included are described in the Appendix A). Of the 65 studies included, 11 were qualitative studies, 1 used mixed methods, and 53 were quantitative studies. Four studies were original or a secondary analysis of a randomized controlled trial. Thirty-two studies extracted data from medical records or administrative databases (Minimum-Data Set (MDS), for example), and 33 studies used different types of surveys (paper- or web-based self-administrated questionnaires, open or directed interviews).

### 3.2. Testing

This category includes nine studies distributed in two subcategories: screening and diagnostic tests [13,14,15,16,17,18,19,20,21].

#### 3.2.1. Screening

Four studies evaluated screening habits in nursing homes for oral, skin, and breast cancers [13,14,15,16].

##### Screening Habits in Nursing Homes

Breast cancer screening habits were evaluated in a study in rural, skilled, long-term care (LTC) facilities of a Midwestern state in the US. It was found that elderly women did not have access to breast cancer screening services within the guidelines of the American Cancer Society and the National Cancer Institute. The majority of respondents estimated that less than 25% of the elderly women in their LTC facility received a clinical breast exam by a provider. The perceived barriers to screening by directors of nursing were that some believed that mammograms were too expensive to be efficient in LTC and facility physician(s) did not encourage mammograms. However, it was reported that the embarrassment for the elderly residents and the time spent for clinical breast exams and mammography would not be a problem to screening services [13].

##### Cancer Screening

An oral cancer screening program in nursing homes in India was implemented and several oral mucosal lesions were found, including leukoplakia in 15.0% of residents and malignant tumors in 1.5% [14]. This illustrates the potential value of screening in the old population of nursing homes. A study conducted in Iran evaluated a similar screening program for dermatological diseases in nursing home residents. With a full-body skin examination by dermatologists followed by appropriate methods to confirm the diagnosis of suspected lesions, it was found that benign neoplasm was the most common disease (68.3%) among patients. None of them had precancerous lesions or skin cancers, which can be explained by a low mean age (73.5 years) [16].

##### Role of the Nurse

The importance of the role of registered nurses in detecting signs of cancer was highlighted in a descriptive survey; from the perspective of 457 nurses, the majority of respondents stated that it was in their scope of practice to teach skin cancer prevention to patients. However, finding skin cancer was a low priority for 62.5% of physicians, and only 60.9% of nurses believed they have a professional role in detection, suggesting that a large proportion of patients will not receive screening measures from their physician or nurse. Nurses also reported many barriers to continuing education in prevention (lack of financial resources, not knowing the educational resources, and time away from home and from work) and detection in nursing home settings (lack of national guidelines, low priority among physicians, and the belief that it is up to the patient to take responsibility for prevention) [15].

#### 3.2.2. Diagnosis

Five studies reported the diagnostic process for cancer in old patients [17,18,19,20,21].

##### Unstaged or Late Disease

A study found that 61.9% of residents were diagnosed late or unstaged, confirmed by high mortality within 3 months of diagnosis (48.4%), and only 27.7% of all residents received hospice care [17]. In Michigan (USA), researchers reported that 25.4% of patients were diagnosed with cancer at death or during the month before their death; 73.9% of diagnoses were made at late stage or were unstaged, and the risk factors for diagnosis at or near death were old age and Alzheimer’s disease [18]. Another study in nursing homes of Ohio (USA) looked at unstaged patients and found that the proportion of unstaged cases was 7.6% for breast, 16.9% for prostate, and 13.8% for colorectal cancer. Patients with higher complexity of care needs were 4 to 5 times more likely to have unstaged cancer [19]. A study conducted in Belgium aimed to describe new cancer events in nursing home residents. Among 4262 residents, only nine had a cancer event, including five new suspected or diagnosed cancer cases, and four cases of progression of a previously known cancer. No diagnostic procedure was performed in four of these nine residents, and no specific cancer treatment was started in five residents. The authors concluded that cancer events that need medical attention occur in nursing home populations at a frequency that is lower than expected, and that therefore, these are underreported, but also that this population does receive diagnostic testing and treatments [21].

##### Physicians’ Motivation for Non-Referral

From the physicians’ perspective, Hamaker et al. explored the motivations behind the choice of breast cancer non-referral by surveying a sample of the members of the National Association of Elderly Care Physicians and Socials Geriatricians in the Netherlands. Of the 419 responders, 32.9% reported not referring the last patient suspected of having breast cancer. Considering that there could be more than one motivation for the same patient, end-stage dementia was cited in 57.0% of cases, patient or family preference in 28.9%, and limited life expectancy in 23.1%. Despite the lack of diagnosis confirmation of breast cancer, physicians felt comfortable initiating hormonal treatment in 16.2% of non-referred patients. When a physician chose to refer a patient for further testing or treatment, the most frequent reasons for referral were to confirm the diagnosis (28.2%), fear of future ulceration or metastases (21.5%), having good general health and life expectancy (19.0%), and the desire of the patient or families to investigate (18.4%) [20].

### 3.3. Treatment

This category includes a total of 12 studies distributed in four subcategories: patient profiles, cancer management in nursing homes, oncological treatment outcomes, and shared medical decisions [17,22,23,24,25,26,27,28,29,30,31,32].

#### 3.3.1. Patient Profiles

Five studies explored the particularities of nursing home residents with cancer and the risk factors for long-term care use [22,23,24,25,26,27].

##### Prevalence and Particularities of Nursing Home Residents with Cancer

Patients diagnosed with cancer represented 7.3% (133) of a sample of 1825 residents in a Norwegian study, of which 100 also suffered from dementia [22]. Buchanan et al. analyzed residents at admission in nursing homes; 11.3% of patients had cancer, 20.7% had a life expectancy of less than 6 months, and 18.7% received hospice care [23]. Residents with cancer felt more pain than those without, more than a half had an unstable health condition, and their general heath was worse [24]. More than half of them needed a wheelchair for locomotion and were dependent for activities of daily living, including walking in a room, dressing, toilet use, and personal hygiene [23]. A study conducted in French elderly nursing home patients described the prevalence of cancer and their functional status; 62.1% of patients were diagnosed before admission. The most common tumor sites were skin, digestive tract, and breast. Staging information was available in only 59.8% of cases, and reports of a multidisciplinary discussion in 34.1%. Exactly 50.0% of subjects were moderately dependent and 43.5% were highly dependent based on a French validated geriatric scale for functional abilities, Autonomie Gérontologique Groupes Iso-Ressources (AGGIR) [25].

##### Risk Factors for Long-Term Care Use

A study evaluated whether a geriatric assessment was predictive of hospitalization and the risk of using long-term care in older patients with cancer. Based on an evaluation between 3 months before and 6 months after cancer diagnosis, they found that predictive factors for LTC use were frailty, difficulty performing instrumental activities of daily living and balance problems revealed by a history of falls, abnormal timed up-and-go test, and limitations in climbing stairs [26].

#### 3.3.2. Cancer Management in Nursing Homes

In the study conducted in France presented above, the decision to undergo a cancer-specific treatment was significantly less frequent among residents with a cancer diagnosed since admission to a nursing home (67.9%) compared to those with a diagnosis made before admission (85.7%, *p* = 0.013). Furthermore, those with a cancer diagnosed during their nursing home stay underwent exclusive symptomatic management more frequently (58.0% versus 25.6%, *p* < 0.001). In cases where it was decided to administer treatment, surgery was the most frequent treatment (55.1%) [25]. A study in the US found that there were few cancer services provided to nursing home patients; 22.3% of residents with cancer (breast, colorectal, lung, or prostate) received cancer-directed surgery (the majority were for breast cancer), and they were younger than those who did not. Only 5.8% of patients received chemotherapy and 6.6% received radiation therapy [17].

#### 3.3.3. Treatment Outcomes

Five studies analyzed the outcomes of common oncological treatments, of which surgery is by far the type of treatment whose outcomes have been studied the most in nursing homes [27,28,29,30,31].

##### Breast Cancer Treatment

A US retrospective study reported that despite increased risk of death from treatment toxicity, adjuvant chemotherapy and tamoxifen significantly improved disease-free and overall survival in older women with node-positive disease. No association was found between age and disease-free survival. The reduction in cancer mortality was equivalent for older and younger women. Overall survival of patients aged 65 years or older was lowered by death from causes other than breast cancer [27]. Of 4960 US nursing home residents who underwent breast cancer surgery, 11.2% had a lumpectomy, 27.5% had a mastectomy, and 61.3% had axillary lymph node dissection (ALND) associated with a lumpectomy or mastectomy. The 30-day mortality ranged from 2.0% to 8.4%, and the 1-year mortality ranged from 29.4% to 41.3% depending on surgery type. The rate of functional decline at 1 year among survivors was 55.3–59.7% and was significantly associated with preoperative functional decline and cognitive impairment [28].

##### Colorectal Cancer Treatment

A study reported that nursing home residents experienced substantial functional decline after surgical resection of colon cancer; the factors associated with this were older age, hospital readmission after surgical hospitalization, surgical complications, and previous functional impairment [29]. A study evaluated the outcomes of this type of surgery in nonagenarians. Postoperative complications occurred in 36.8%, and pneumonia was the most common complication. The 30-day mortality rate was 7.0%, and the 180-day mortality rate was 31.6%. The authors concluded that outcomes could have been more favorable in a selected group of patients [30]. Another study evaluated mortality and bowel function after proctectomy in residents. Operative mortality was 17.6% in cases of permanent colostomy and 13.3% in cases of sphincter-sparing proctectomy; fecal incontinence was reported in 37.3% of cases, and the risk factors of this were poor functional status, renal failure, and dementia [31].

#### 3.3.4. Shared Medical Decision

To understand how family members weighed possible outcomes of medical decisions with or on behalf of nursing home residents with cancer, a qualitative study purposed to use the prospect theory’s concepts of gain, loss, risk, and reference point. Three themes highlighted by participants of this study included “Don’t prolong this”, “A good ending is a gain”, and “Experience can facilitate seeing the big picture”. Compromising the chance for a peaceful dying process was considered unacceptable for family members [32].

### 3.4. Symptoms

This category includes 16 studies distributed in six subcategories: physical symptoms, impact of cognitive impairment, pain management, severity of pain, neuropsychiatric symptoms, and the role of the nurse in psychological symptoms [22,33,34,35,36,37,38,39,40,41,42,43,44,45,46,47,48].

#### 3.4.1. Physical Symptoms

A study compared symptoms in residents with and without cancer in nursing homes in the state of Kansas, USA. Residents with cancer had significantly more frequent pain, shortness of breath, vomiting, weight loss, and diarrhea [33].

#### 3.4.2. Impacts of Cognitive Impairment

Three articles found a negative correlation between cognitive function and total amount of opioid medication, indicating that severely demented patients received less opioids [30,31,32]. From another point of view, among patients with dementia, those diagnosed with concurrent cancer received more analgesic medication than those without cancer; they also had significantly more neuropsychiatric symptoms, particularly sleep disturbances and agitation, suggesting that symptoms may not be treated adequately [22]. Monroe et al. reported that 40.0% of nursing home residents with dementia who died from cancer were not receiving any opioid when they died, and although patients who benefited from hospice care were more likely to receive an opioid analgesic during the last 2 weeks of life compared to those who did not, they were also less likely to show severe cognitive impairment [37].

#### 3.4.3. Pain Management

A group of researchers in Norway found that the diagnosis of cancer led to an increase in the use of pain relievers over time, reflecting that cancer patients were in more pain than those without cancer. The main indication for using opioids close to death in nursing home residents, regardless of cancer diagnosis, was pain (22.0%), a combination of different reasons (22.0%), prophylaxis (18.0%), dyspnea (14.0%), sedation (11.0%), or for comfort (13.0%) [38]. When a patient received palliative care, pain management was enhanced with scheduled and as-needed medication [39]. Co-analgesics, generally added to maximize the effectiveness of opioids, have been studied in nursing home residents with cancer and were present in the pharmacological profile of 14.2% of admissions. Factors associated with this type of treatment were advanced age and comorbidities (including dementia and depression). Gabapentin, a gamma-aminobutyric acid (GABA) analog, was the most common adjuvant drug [40].

#### 3.4.4. Severity of Pain

A study evaluated the intensity of pain in nursing homes in the USA. Among residents, 3.7% had daily pain that was excruciating at least once in the previous week; this was reported more frequently in younger patients, and more than 20% were diagnosed with cancer. Among residents with this level of pain, 62.1% needed help taking care of themselves and 48.8% had normal cognitive status [41]. Another study reported that 65.6% of residents with cancer suffered from pain, the severity of which was severe for 13.5% of them and moderate for 61.3%. Pain treatment was significantly and negatively associated with age greater than 85 years, cognitive impairment, and the presence of a feeding tube and of restraints [42]. From the perspective of family members of nursing home residents with terminal cancer, those who believed that better pain management was possible became the patient’s advocate and ensured that this was addressed [43].

#### 3.4.5. Neuropsychiatric Symptoms

Drageset et al. conducted a study investigating loneliness among cognitively intact patients with cancer living in nursing homes. Loneliness was a common psychosocial effect after cancer diagnosis. It was described as a feeling of inner pain, loss, and inferiority. This emotional state was alleviated by being engaged in activities, being in contact with other people, and by taking care of oneself [44]. The same authors found that emotional loneliness, age, education, and comorbidity influenced mortality among nursing homes residents without cognitive impairment, irrespective of cancer diagnosis [45]. In another study, the authors also found that anxiety symptoms according to the Hospital Anxiety and Depression Scale were associated with worse survival compared to the absence of symptoms in residents with cancer; this association was not found in the group without cancer [46].

#### 3.4.6. Role of the Nurse in Psychological Symptoms

A study analyzed the communication patterns between patients, their family, and nurses in a cancer home hospice setting. When a high level of distress was felt by the patient, the role of the nurses was to respond by providing information, offering partnership, and being emotionally responsive [47]. In another study, a patient education program delivered by nurses decreased the perception of fatigue in patients with gastrointestinal cancer [48].

### 3.5. Advanced Care Planning (ACP)

This category includes a total of seven studies distributed in three subcategories: do-not-resuscitate (DNR) wishes, trajectory of the disease as a tool for ACP discussions, and implication of nurses in decisions about goals of care [49,50,51,52,53,54,55,56,57].

#### 3.5.1. Do-Not-Resuscitate Wishes

Only 11.3% of residents from a nursing home in northern Taiwan had a signed DNR will or consent form [49], while DNR wishes are documented in 39.4% of nursing home residents in US [57], indicating that there are cultural differences in end-of-life decisions. Severe brain injury, pulmonary disease, and cancer were all significant triggers to initiate DNR discussions. Cancer residents had a higher likelihood of completing DNR forms compared to non-cancer residents, which might be explained by a palliative goal of care for a better quality of life [49]. A study conducted in multiple healthcare centers in Henan, China, aimed to examine the attitudes of patients with cancer and health professionals towards palliative care. Even if death was accepted as a natural life process, 43.0% of patients believed life preservation was their major goal of care. The importance of palliative care was recognized by 87.9% of doctors and 82.1% of nurses, but less than half (48.8%) were trained for or were willing to deliver the relevant care [52].

#### 3.5.2. Trajectory of Disease as Guide for ACP

A study identified the trajectory of functional decline based on different diagnoses as a predictive tool for planning ACP in a nursing home population. Cancer patients and frail residents both had significant curvilinear terms. Their level of function declined by 15.9% and 20.6%, respectively, from admission to death; the cancer group had a shorter mean length of stay (2.2 years, 95%CI [1.3; 3.1]) than the frail group (3.4 years, 95%CI [2.9; 3.9]) [53].

#### 3.5.3. Implication of Nurses

A study evaluated the feasibility and acceptability of nurse-led health-related value discussions with new cancer patients. It appeared to be helpful from the patients’ point of view. From professionals’ perspective, normalizing these discussions at the time of diagnosis was an improvement of care quality and allowed to establish a framework for shared decision-making before a progression, a complication, or an end-of-life context [54]. Embedding a palliative care nurse practitioner for patients with advanced cancer has been shown to increase the likelihood of documenting ACP and receiving referral for psychosocial support [55]. Another study demonstrated that the role of the nurse was important for decisions concerning the intensity of therapy. It was reported that less than half of residents were documented as participating in the conversation, and even fewer when they had dementia and lived in long-term care. In these cases, nurses participated in approximatively 67% of the discussions, while physicians did so in 34% [56].

### 3.6. End of Life

This category includes nine studies distributed in two subcategories: mortality of cancer in nursing homes and caring [58,59,60,61,62,63,64,65,66].

#### 3.6.1. Places for Death

According to a population-based study conducted in Nova Scotia, Canada, breast cancer decedents were more likely to die in a nursing home compared to all-cancer causes and were twice as likely to have dementia as a cause of death [58]. In a study conducted by Bainbridge et al., 61.3% of residents had visited an emergency department in the last 6 months of life and 20.4% had died in hospital. The utilization of acute care by nursing home residents dying of cancer reflected an unstable disease or uncontrolled symptoms [59]. In another study, the hospice utilization rate reported was 52.4%, with 70.8% for cancer deaths and 45.4% for non-cancer deaths in US nursing homes. Cancer was associated with higher odds of receiving hospice care [60]. A study aimed to compare trajectories before death in long-term care for patients with cancer, dementia, or chronic illness. Diagnoses at the time of death were dementia (49.0%), chronic illness (30.0%), cancer (17.0%), and dementia combined with cancer (4.0%). People with dementia had more pain and physical symptoms, and those with cancer had less anxiety [61], contrary to the findings mentioned in previous sections. Drageset et al. evaluated mortality in residents without cognitive impairment and its relation to self-reported health-related quality of life (HRQoL) and other factors. Having a cancer diagnosis did not predict mortality, but poorer physical functioning was significantly associated [62]. Another study reported that cancer was more strongly associated with mortality in women than in men in newly admitted residents [63].

#### 3.6.2. Caring

A qualitative case study about the experience of five registered nurses caring for a person dying of cancer in a nursing home revealed four major particularities of their practice: exclusivity of the relationship with the patient, difficulties in the management of pain, expectations of the nurse along with the influence of staff, and time constraints and the impact of caring [64]. A study in Norway assessed barriers to or facilitators for improving palliative care for patients with cancer and dementia from the perspective of professionals. The interviews revealed three specific barriers to palliative care: assessment tools often being inappropriate for this type of patient, tension felt when changes are at odds with the holistic hospice care philosophy (experiential knowledge of nurses confronted by a purely medical focus), and lack of expertise [65]. Another group of researchers highlighted five factors that affect the quality of palliative care: communication difficulties between services and between professionals and patients and their families; variable extent of structural/functional integration of services; difficulties in funding palliative care services; problematic processes of care, including definition and misperceptions of palliative care; lack of knowledge in dementia and end-of-life care; and time constraints. Communication with people with terminal dementia was a challenge because they often had less ability to communicate verbally and it affected the assessment of needs and subsequent care [66].

### 3.7. Nursing Home Care

This category includes seven studies distributed in three subcategories: deprescription, difficulties related to transition between long-term care and hospitals, and quality indicators [67,68,69,70,71,72,73].

#### 3.7.1. Deprescription

A study in Chicago, US, aimed to evaluate the continuation of limited-benefit medication (LBM) following admission of residents with cancer in nursing homes. At least one LBM was continued in 29.8% of patients with cancer and in 30.5% of patients with a non-cancer-related cause. The most frequently continued medications were anti-dementia (29.3%), and antiosteoporosis medications were least often maintained (14.1%). When admitted in skilled nursing homes, LBM continuation was greater compared to that in home hospices (RR 1.25, 95%CI [1.20; 1.29]), non-skilled nursing homes (RR 1.29, 95%CI [1.25; 1.32]), and assisted living facilities (RR 1.28, 95%CI [1.24; 1.32]) [67]. The SHELTER study assessed 1-year incidence and factors related to deprescribing in nursing home residents with polypharmacy in Europe. They found that the mean number of medications was 8.6 at the baseline, and deprescribing was observed in 35.6% of residents. The factors with the highest probability of reducing the numbers of drugs were cognitive impairment, presence of a geriatrician within the facility staff, and number of medications used at baseline. The factors associated with a lower probability of deprescribing were cancer, heart failure, and female gender [68].

#### 3.7.2. Transitions between Acute and Long-Term Care

A cohort study in Norway observed that residents with cancer had 1.7 times higher risk for acute hospital admissions than those without [69]. Another study examined factors associated with potentially burdensome end-of-life transitions (two or more hospitalizations or an intensive care unit admission in the last 90 days of life) between settings among elderly residents of US nursing homes with poor-prognosis cancer. Thirty-six percent of patients experienced a potentially burdensome end-of-life transition, and it was more common among patients who did not receive hospice care (45.4% compared to 28.7%). Associated factors were full dependence on ADLs, congestive heart failure, and chronic obstructive pulmonary disease. Those with impaired cognition and do-not-resuscitate orders had the lowest risk of transfer [70]. A study explored frontline nurses’ perspectives on the transition of cancer patients with care needs from a gastroenterology ward. Optimal follow-up care after discharge of these patients was influenced by the complexity and fluctuating health status. Determining factors considered in order to avoid readmission to the surgical ward when planning discharge to a nursing home were age, physical condition, nurses’ determination of their pain level, and competency in palliative care [71].

#### 3.7.3. Quality Indicators

A study compared and correlated the experiences and satisfaction of caregivers as quality indicators between two diagnoses: cancer and heart failure. Nursing home placement was more than twice as likely to happen to heart failure patients compared to cancer patients and was a factor of dissatisfaction with care in both groups [72]. Nutrition status was studied as a common problem in healthcare settings. The prevalence of malnutrition was 19.2% in nursing homes, 23.8% in hospitals, and 21.7% in home-care organizations. Dementia was associated with malnutrition in nursing homes, while cancer was one of the associated factors in home care [73].

### 3.8. Education

This category includes seven studies distributed in two subcategories: facilitators of continuing education and educational strategies [65,74,75,76,77,78,79].

#### 3.8.1. Facilitators of Continuing Education

Health professionals were interviewed in a study in Norway and proposed examples of educational strategies and facilitators: lectures, workshops, formal meetings, reminders with laminated instruction cards in plain sight, and a person with special training that can tutor colleagues on workdays. Facilitators that increased the educational impact were repeated trainings, qualified as mandatory, accessible for part-time staff, and half- or full-day courses preferred over short sessions [65]. Another study explored nurses’ perceptions about recruitment, retention, and workplace strategies in oncology. Continuing education opportunities were considered critical to job satisfaction at the time. Four strategies were reported to facilitate continuing education: recognition of this field as a specialty, recognition that tacit knowledge is no longer enough, gratification as a retaining factor, and acknowledgement that the relationship depends on the environment. If cancer nursing was considered not only as the practice of administration of chemotherapy but more as a combination of clinical practice experiences in an inpatient setting and life experiences necessary to deal with life-altering cancers, continuing education would be greatly facilitated [74].

#### 3.8.2. Educational Strategies: Training Programs and Problem-Based Learning

##### Training Programs

A randomized controlled study evaluated an educational program for nurses about caring for patients with malignant pleural mesothelioma in Japan. The 2-day program of 14.5 h included lectures on epidemiology, diagnosis, chemotherapy, surgery, and patients’ needs, group work, role-playing, and group discussions. The knowledge, difficulty, and attitude scores were significantly more favorable with the intervention post-test and follow-up test 1 month later [75]. Gerhart et al. conducted a study that assessed a brief provider training program’s feasibility to increase self-efficacy in responding to patient anger. A communication training program was conducted over the course of 1 h, mid-day, in-service by a licensed clinical psychologist and a postdoctoral fellow in clinical health psychology. It included discussions following a video, didactics, a role-play session, and a stress management experience. They observed a large improvement in 9/10 skill outcomes, including acknowledging patient anger, discussing anger, considering solutions, and using relaxation to manage their own distress [76]. Another study evaluated the recognition of depressive disorders in cancer patients after a brief 1-h didactic training session for oncologists and nurses. The program included a didactic lecture, videotaped interviews, and case discussions. Based on the evaluation of two cases, there was a high concordance among staff regarding symptom ratings. The authors concluded that they were able to identify depressive symptoms in cancer patients on videotape [77]. As ineffective communication might leave patients feeling anxious and dissatisfied, a study investigated if a communication skills training program for nurses in breaking bad news would be useful for improving the quality of life of patients and their satisfaction with healthcare professionals at the time of diagnosis. The training program consisted of two workshops, one at the start and the other after 3 months, lasting 6 h each. The program was based on a six-step protocol, referred to by the acronym SPIKES (developed by Baile et al. and Fujimori et al. [80,81,82]): setting up the interview, assessing the patient’s perception, obtaining the patient’s invitation, giving knowledge and information, addressing the patient’s emotions with empathy, strategy, and summary. The results showed a marginal improvement in mental aspects of quality of life with time and satisfaction with nurses [78].

##### Problem-Based Learning

A study developed two problem-based learning packages in cardio-pulmonary medicine including clinical cases and tutorial guidelines. Among participants, 57.5% found that problem-based learning in continuing nurse education was a strong motivator for self-learning and promoted active attitudes in cooperative learning. On the other side, 20.0% of the participants answered that the PBL method was not suitable for clinical nurses and represented a significant burden in preparing individual learning tasks compared with lecture-based education [79].

## 4. Discussion

This review highlights the multidimensional complexity and the possible organizational, cultural, and ethical barriers of care, as well as mapping the educational strategies for paramedical staff working in nursing homes and taking care of cancer patients (Figure 2).

The end of cancer screening programs after 75 years of age does not mean the end of the cancer risk in this population [1,2,3,83]. Despite the high prevalence of this disease in the elderly, early diagnosis appears to be neglected in nursing homes compared to screening guidelines [13]. The potential explanatory factors for this phenomenon are found in the lack of willingness of nursing home physicians to carry out the necessary tests [13] and the non-perception by nursing staff of their potential role in identifying signs of cancer during daily care [15]. The complexity of organizing tests and appointments for patients living in this particular setting may contribute to the tendency to refer them less frequently for further testing, but no study addressing this topic was found in our search. Therefore, once cancer is diagnosed, elderly patients living in nursing homes are at higher risk of suffering from a late or even unstaged disease [17,18,19,20].

It is known that certain treatments against cancer in frail patients can lead to functional decline and excess mortality [27,28,29,30]. However, there was no published study found on the outcomes and feasibility of chemotherapeutic treatment in nursing home residents, or specifically focused on oral chemotherapy in a nursing setting. There are also no data concerning the barriers perceived by nurses regarding the management, handling, and monitoring of the toxicity of this treatment. When a cancer is diagnosed during a nursing home stay, the therapeutic strategy leans more towards comfort care rather than active and specific cancer care compared to other care settings [17,25]. Concomitant dementia also influences the management of the disease and symptoms [22,33,34,35,36,37,38,39]. Agism, in part due to a lack of knowledge, is a cultural barrier that limits the access of older adults to specific cancer care, which is considered as care that is hostile, a source of deteriorating quality of life, and therapeutic obstinacy. However, older patients do accept cancer therapy as well as younger patients but are not willing to trade survival for current quality of life [4]; it should be noted that these treatments may have benefits in terms of quality of life and management of pain and complications if the risk of adverse effects associated with the treatments is considered acceptable according to the general state of health of the patient [27]. Conversely, it is unethical to suggest or recommend treatments that have no chance of physiological or clinical benefit, which would correspond to the concepts of medical futility [84] and potentially inappropriate treatment [85]. This therefore explains the importance of thoroughly evaluating frailty and the individual risk of toxicity, balanced by the potential benefit of a treatment [86,87]. In parallel, decision processes about end-of-life care, ACP, and DNR may vary according to the culture of the population and the expertise of health professionals [49,51,52,57]. In this context, nurses are a key member of the healthcare team that provide information, explore the values of patients, and help with decisions regarding the intensity of care and DNR [54,55,56].

Cancer is a frequent cause of death in nursing homes [58,61,63], although death is more related to functional decline than to tumor cancer itself [62]. When approaching the end of life, caring for a dying patient is known to be challenging [64,65,66]. Comfort care and deprescription as standards of care in nursing homes are sometimes in conflict with the continuum of cancer care that has the potential to vary in therapeutic intensity and medical instability over time [67,68,69,70]. Incapacity to achieve comfort and manage medical instability in this setting may lead to acute care transfer and, therefore, care discontinuation because of bidirectional communication difficulties [69,88].

There is a strong need for continuing education felt by nurses [74]. Each category of barrier highlighted in the current study represents barriers to optimal cancer care in nursing homes and, therefore, potential targets for continuing the education of nurse practitioners. Unfortunately there is little evidence on the learning devices that could be the most effective, accepted, and implementable [75,76,77,78,79]. Many education tools have been studied in nursing students but not in registered nurses, such as microlearning [89], social media [90], and lifelong learning [91]. There is, therefore, an important need for research in this area.

Nursing care dominates nursing home practice [92]. They are at the center of the continuum of care for patients living in nursing homes and, therefore, the main target of the ONCOPAD program. The current study is the first step of this program. Co-construction of educational tools will be dedicated to paramedical staff in nursing homes, intended to demystify the management of certain cancers (breast, prostate, etc.) whose risk–benefit balance remains favorable, to develop their knowledge and skills to improve diagnostic and therapeutic decision-making, and to improve the consistency of care and, ultimately, quality of life.

This review has revealed a lack of evidence concerning cancer treatment in nursing homes. It was, therefore, necessary to use a broad research question in order to indirectly answer our hypotheses. Several limitations should also be noted. The selected studies had some methodological weaknesses; for instance, a large majority of the studies were descriptive, single-center, and small in size. Furthermore, most data from prospective studies came from open or semi-structured interviews, and few were based on validated questionnaires. For this reason, the results were hard to compare, and the generalizability was uncertain. Furthermore, it was difficult to draw conclusions from studies that had heterogeneous definitions of age groups, nursing homes, functional reservation and basal health conditions. This review still has some strengths. Despite the diversity of studies, several obstacles to cancer care in nursing homes have been defined in every step of the continuum of care.

## 5. Conclusions

Despite the proven benefits for elderly cancer patients and caregivers of early diagnosis and appropriate therapeutic management across the continuum of care, in terms of quality of life or care complexity, these appear to be comparatively less well developed in nursing homes and are subject to multiple barriers. To optimize care for older patients and to combat fear and resistance to cancer care among nursing home staff, there is an important need to educate nurses about recognizing and managing the signs and symptoms of cancer in elderly people living in nursing homes. There is also a strong need for research on cancer treatment options in residents and their outcomes, and for continuing education strategies for nurses.

## Figures and Tables

**Figure 1 cancers-13-03514-f001:**
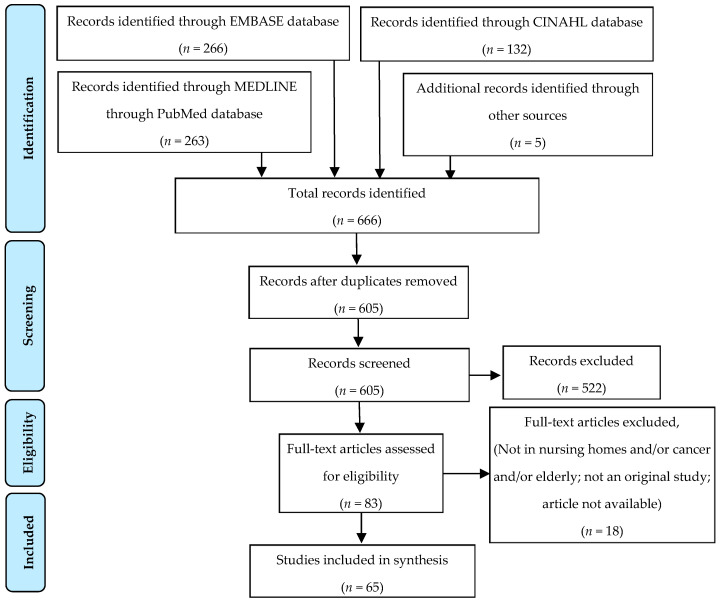
PRISMA flowchart summarizing the process for the identification of eligible articles.

**Figure 2 cancers-13-03514-f002:**
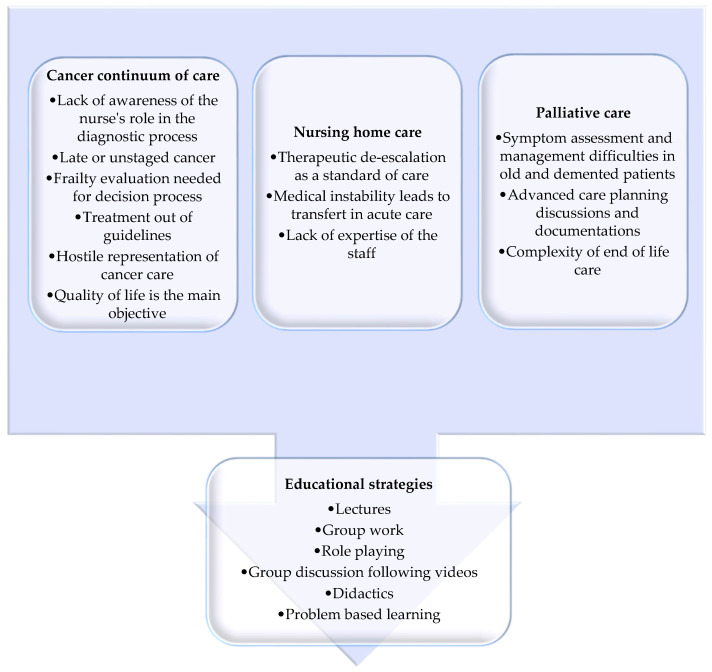
Main barriers of cancer treatment in nursing homes divided in cancer continuum of care, palliative care, and nursing home care, as well as the solutions represented by educational strategies.

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
