# Peer review of "Cultural and Ethical Barriers to Cancer Treatment in Nursing Homes and Educational Strategies: A Scoping Review"

_cancers, 2021, doi:10.3390/cancers13143514_

Round 1

Reviewer 1 Report

Thank you for giving me the opportunity to review the manuscript entitled “Cultural and Ethical Barriers to Cancer Treatment in Nursing Homes and Educational Strategies: A Scoping Review“. Over the past decade, a large body of evidence has supported enhancing the quality of life at different stages of disease. The questions is no longer whether best possible care should be offered, but more how it can appropriately be provided. Therefore, concerning the current manuscript comprising a scoping review about cancer treatment in nursing homes, the authors are to be congratulated for conducting rigorous research. The manuscript is easy to follow and does have a logical flow. It is written straight forward and focuses on an interesting topic of high relevance. The title and abstract cover the main aspects of the work. The current evidence and the statistical methods are well described, the study results are well presented and the manuscript does provide an advance in the field of oncology. It was a pleasure to read the manuscript. Minor comment: Could the authors please provide a statement about the term of ”futility” as this seems relevant within this context.

Author Response

Dear reviewer,

Please find appended the revised version of the manuscript entitled “Cultural and Ethical Barriers to Cancer Treatment in Nursing Homes and Educational Strategies: A Scoping Review”, and below a point-by-point reply to comments made.

Thank you for agreeing to review this manuscript and for the constructive comments.

We hope that this version is acceptable for publication but remain at your disposal if any further modifications are required.

Thank you for your consideration,

Yours faithfully,

Cynthia Filteau, on behalf of the co-authors.

Point 1: Could the authors please provide a statement about the term of “futility” as this seems relevant within this context.

 Response 1: Following your suggestion, we have clarified the concepts of futility and potentially inappropriate treatments where the risks are expected to outweigh the potential benefits of improving quality of life.

Reviewer 2 Report

It was a pleasure to review this scoping review of a very important and under-researched topic on the barriers to providing cancer treatment in nursing homes.  Your manuscript is well-written and the flow of the barriers and opportunities identified from the literature reviewed was easy to follow. 

Title: appropriate

Abstract: appropriate

Purpose: appropriately stated

Methods: appropriately outlined following PRISMA-Sc guidelines; included 3 databases for relevant literature and one database for gray literature.  Inclusion/exclusion criteria for selections; review process; extraction process addressed.

Results: Appropriate concise descriptions of articles addressing specified topics for barriers, nursing home care delivery, educational strategies for this vulnerable population.  Adhered to the definitions that were provided to guide the reader.

Discussion: robust with many gaps in the literature addressed; considerations for overcoming barriers to care and educational training needed to promote adequate cancer care provisions to this population in nursing homes

Conclusions: appropriate but need to include symptom recognition and management as an additional focus for training

Tables: appropriate

References: appropriate

Author Response

Dear reviewer,

Please find appended the revised version of the manuscript entitled “Cultural and Ethical Barriers to Cancer Treatment in Nursing Homes and Educational Strategies: A Scoping Review”, and below a point-by-point reply to comments made.

Thank you for agreeing to review this manuscript and for the constructive comments.

We hope that this version is acceptable for publication but remain at your disposal if any further modifications are required.

Thank you for your consideration,

Yours faithfully,

Cynthia Filteau, on behalf of the co-authors.

Point 1: Need to include symptom recognition and management as an additional focus for training.

Response 1: Following your suggestion, we have revised the manuscript to reiterate in the conclusion the need to educate nurses about recognition and management of signs and symptoms of cancers.
